# Improved Visible Emission from ZnO Nanoparticles Synthesized via the Co-Precipitation Method

**DOI:** 10.3390/ma16155400

**Published:** 2023-08-01

**Authors:** Alexandra Apostoluk, Yao Zhu, Pierrick Gautier, Audrey Valette, Jean-Marie Bluet, Thibaut Cornier, Bruno Masenelli, Stephane Daniele

**Affiliations:** 1Université de Lyon, INL-INSA Lyon, CNRS, UMR 5270, 69621 Villeurbanne, France; aleksandra.apostoluk@insa-lyon.fr (A.A.);; 2Université de Lyon, IRCE Lyon, CNRS, UMR 5256, 69626 Villeurbanne, France; 3C2P2-UMR 5265, ESCPE-Lyon, BP 2077, Univ Lyon, Université Claude Bernard Lyon 1, 69616 Villeurbanne, France

**Keywords:** ZnO nanoparticles, co-precipitation, luminescence quantum efficiency

## Abstract

Since ZnO nanoparticles (NPs) possess a variety of intrinsic defects, they can provide a wide spectrum of visible emission, without adding any impurity or any doping atoms. They are attracting more and more interest as a material for light sources and energy downshifting systems. However, defect emission with a high luminescence quantum efficiency (PL QY) is difficult to obtain. Here, we present the co-precipitation synthesis parameters permitting to attain ZnO NPs with highly visible PL QYs. We found that the nature of zinc precursors and alkaline hydroxide (KOH or LiOH) used in this method affects the emission spectra and the PL QY of the as-grown ZnO NPs. LiOH is found to have an advantageous effect on the visible emission efficiency when added during the synthesis of the ZnO NPs. More precisely, LiOH permits to increase the emission efficiency in the visible up to 13%. We discuss the effects of the nanoparticle size, the morphology and the surface stabilization on the enhancement of the luminescent emission efficiency. Various spectral contributions to the luminescent emission were also examined, in order to achieve a control of the defect emission to increase its efficiency.

## 1. Introduction

ZnO is a II-VI group direct wide bandgap (3.37 eV at room temperature) semiconducting material [1], which has usually been used for applications in the UV spectral range [2,3]. A good crystalline quality of ZnO is required to obtain high efficiency and high sensitivity in UV devices. The recombination on the inevitable intrinsic defects in ZnO is competing with the excitonic emission, making the efficiency of the UV devices low. Much effort has been devoted to reducing these undesirable defects [4,5]. However, currently the intrinsic defects in ZnO are attracting more and more interest as they give rise to a wide range of light emission from blue to IR [6,7], especially in the ZnO nanostructures. The latter are expected to possess a high concentration of defects as they have high surface-to-volume ratios. When defects are engineered accordingly and their presence does not result in the increase of the non-radiative desexcitation, the color of the defect emission of ZnO nanostructures can be tuned easily and the luminescent efficiency can be dramatically enhanced. These nanostructures are promising materials as energy downshifting systems for the application in solar cells [8] and white light sources [9] and can act as an active photocatalyst under visible light illumination [10].

ZnO nanostructures have been synthesized using methods such as the low energy cluster beam deposition (LECBD) [11], ultrasonic synthesis [12], sol–gel synthesis [13] and hydrothermal method [14]. It occurs that the synthesis conditions have a strong effect on the defect emission, and it is challenging to control this emission accurately. Furthermore, the precise mechanisms of these defect emissions remain controversial still [15,16], making it more complex to achieve a high luminescent emission efficiency related to defects.

The co-precipitation method is a widely used sol–gel synthesis technique, as it necessitates conditions such as an ambient environment and room temperature and is cost-effective. Moreover, the synthesis conditions can be easily controlled, and different surfactants can be introduced to keep the as-prepared nanoparticles (NPs) unchanged. There have been many reports on the ZnO nanostructures synthesized by the co-precipitation method [17,18]. The solvent and the nature and quantity of alkaline cations are viewed as the most important parameters which affect the nanostructure size and thus influence the defect emissions [19,20]. LiOH usually induces a smaller size of ZnO NPs compared to KOH or NaOH [19]; however, the influence of the synthesis parameters is rarely correlated to the nature of the emission center (defect’s nature) in the final ZnO nanostructures. In this work, ZnO nanoparticles have been synthesized via the co-precipitation method and the influence of the synthesis conditions has been studied in depth in its various facets, such as the nature of the precursor, the type and the quantity of alkaline hydroxide and the reaction solvent’s effect on the nanoparticle crystalline quality, size, morphology and optical properties [21,22].

## 2. Materials and Methods

### 2.1. Preparation of ZnO Nanoparticles

In the co-precipitation synthesis method, 20 mmol of zinc acetate dihydrate (Zn(OAc)_2_·2H_2_O) dissolved in 70 mL of ethanol or water were simultaneously precipitated using an aqueous or ethanol solution containing 80 mmol of KOH. The reaction medium was stirred for 1 h and then centrifuged for 10 min, in order to isolate the synthesized solid ZnO. The obtained powder was washed with 20 mL of water and then with 20 mL of ethanol. Then, the white solid was dried in an oven at 70 °C for 4 h. For some syntheses, the precursor Zn(OAc)_2_·2H_2_O was replaced by zinc sulfate (ZnSO_4_) and KOH was replaced with varied quantities of LiOH: the ratios of the volume of LiOH solution to the zinc acetate being 2:1 (2 eq.) or 4:1 (4 eq.), in order to analyze the effect of the alkaline ion and of the precursor on the photoluminescent emission of the ZnO NPs.

### 2.2. Characterization of the ZnO Nanoparticles

The crystalline structure was investigated by Raman spectroscopy (Raman Thermo-Fisher DXR, Waltham, MA, USA) using the laser excitation at 532 nm with the intensity of 10 mW. A Bruker (Siemens, Billerica, MA, USA) D5005 diffractometer using the K-alpha radiation of Cu (1.54184 Å) permitted us to roughly estimate the size of the nanoparticles from X-ray diffraction (XRD) analysis using the Debye–Scherrer formula. The morphology of the nanoparticles was analyzed using Transmission Electronic Microscopy (TEM JEOL 2010F operating at 200 kV, Tokyo, Japan).

The samples were analyzed with the Nuclear Reaction Analysis (NRA) method to study the chemical composition of ZnO NPs. The nuclear reaction analysis (NRA) is a technique for the detection of the concentration as well as the depth profile of light chemical elements present in a heavy matrix. When a particle of high energy (protons or α particles) hits a nucleus of an element, then depending on the mass and the energy of the incident particle and of the target element mass, a diversity of final products may be obtained. In cases where the incident beam energy exceeds a certain threshold value, energetic particles appear in the spectrum (non-elastic scattering). The detection of these particles usually provides information about the chemical elements present in the sample. The content of lithium (Li) and potassium (K) in the ZnO NPs was investigated using the nuclear reaction of ^7^Li (p, α) α. An incident proton beam of the energy of 1.5 MeV was used and the quantity of the formed α particles was measured to determine the Li and K content in the ZnO NPs. The minimal element concentrations than can be detected with this method is 0.5%. Elemental analysis of Li content was also determined by the “Service d’Analyse” of IRCELYON.

The optical properties of the nanoparticle assemblies were measured using the photoluminescence (PL) method. A continuous laser emitting at 266 nm (power of 1.2 mW, Crylas FQCW 266-10 from Crylas Laser Systems, GmbH, Berlin, Germany) was used as an excitation source. The PL emission of ZnO nanoparticles was focused with a lens onto an optic fiber, dispersed by an iHR Triax 320 Jobin-Yvon, from Horiba Jobin Yvon, Longjumeau, France, spectrometer and detected using a Si CCD detector (Horiba Jobin Yvon, Longjumeau, France) cooled with liquid nitrogen. As the main objective of this work is to fabricate nanoparticles with the highest PL yield possible, the photoluminescence quantum yield (PL QY) of our samples was measured using an integrating sphere. For the PL QY measurements, the same excitation and detection set-up was used as for the PL study. The final PL QY of the studied samples was calculated according to the procedure proposed by De Mello et al. [23]. For both PL and PL QY measurements, the indium foil was used as a substrate, onto which the ZnO nanoparticle powders were deposited.

## 3. Results and Discussion

### 3.1. Influence of the Solvent Used in the Co-Precipitation Method

The first parameter controlling the co-precipitation method is the nature of the solvent. The solvent influence on the final ZnO NPs was investigated in function of its polarity. Therefore, water and ethanol were used, as they are, respectively, polar and non-polar solvents. It has been proven that the polarity of the reaction medium influences the size of the final nanoparticles. Hexagonal wurtzite ZnO nanoparticles (JCPDS file no. 04-003-2106) were obtained in both cases, which can be seen in the XRD diffractograms presented in Appendix A. It can also be observed that the XRD peaks’ widths of ZnO NPs synthesized in ethanol are broader than that in water, indicating the mean NPs’ size is smaller in ethanol (about 10 nm) than in water (around 20 nm), as roughly deduced from the XRD diffractograms. The PL spectra of ZnO NPs synthesized in water and in ethanol result in the similar spectrum in the visible peaking at around 610–620 nm, as can be seen in Figure 1, though zinc NPs synthesized in water give a slightly wider visible emission than these synthesized in ethanol and a second emission peak, at 755 nm, can be observed, attributed to the radiative recombination of shallowly trapped electrons with deeply trapped holes at oxygen interstitials [24]. Furthermore, similar ratios of visible to UV emission intensity (marked as VIS/UV in Figure 1) and PL QY of about 2% for both types of ZnO NPs are obtained.

Since water is a solvent that is cheaper and less hazardous to humans and to the environment and, as far as the industrial applications are concerned, the solvent nature does not make a big difference in the PL emission (see Figure 1), water has been chosen to synthesize the nanoparticles, with the purpose of studying the influence of the alkaline ion (LiOH or KOH) and the zinc precursor (acetate or sulfate) on the properties of final ZnO nanoparticles.

### 3.2. Crystalline Structure of ZnO Nanoparticles

To study the effect of the alkaline hydroxide on the ZnO NPs, four equivalents of both types of alkaline hydroxide (KOH and LiOH) with respect to both zinc precursors (acetate and sulfate) were used in the co-precipitation synthesis. The XRD diffractograms presented in Figure 2 demonstrate that with both zinc precursors and with both alkaline hydroxide solutions, wurtzite ZnO NPs of a good crystalline quality have been obtained. However, minor additional peaks can be seen in the spectra of ZnO NPs synthesized with LiOH (bottom curves in Figure 2a,b), attributed to the presence of the tetragonal sweetite phase Zn(OH)_2_ (JCPDS file no. 00-38-0356). The ratio between these two phases could not be quantified, since there exists only one file for the sweetite phase and none of the established structure. It can be seen in Figure 2 that when KOH is used for both precursors, the full width at half maximum of the XRD peaks of (002) planes is smaller than that of the other peaks. When LiOH is used, all the peaks have almost the same width. Concomitantly, the (002) diffraction peak intensity is larger when KOH is used as compared to the case when LiOH is used. In the latter case, the intensities of the different peaks correspond to the ones expected from powder diffraction. These observations show that KOH induces a preferential growth along the [002] axis, as was already reported by Spanhel et al. [25]. It can also be clearly seen that as KOH is replaced with LiOH, the XRD peaks’ widths increase when zinc acetate is used while they decrease when zinc sulfate is used. It indicates that as the nature of alkaline ions changes, a difference in size variation trend occurs when different precursors are used. The sizes of the crystalline domains of the studied nanoparticles are given in Table 1. They were estimated from the XRD diffractograms, using the Scherrer formula:(1)$$\tau =\frac{K\lambda }{\beta \cos\theta }$$
where *τ* is the mean size of crystalline domains, *K* is a constant with a typical value of 0.9, *λ* is the wavelength of the X-rays, *β* is the full width at half maximum (FWHM) of the diffraction peak intensity and *θ* is the incident angle of the X-ray beam.

The values of the crystalline domain sizes obtained from the calculations using the FWHM of the (002) peak give values higher than (101) for the case of potassium hydroxide, as can be seen in Table 2. It can be noticed in Figure 2 that the (002) peak is more intense in comparison to the (101) peak in the case of lithium hydroxide used in the same proportions. It means that the ZnO NPs grow preferentially in this direction when the KOH is used. It can be also confirmed by the TEM images presented in Figure 3b,d, where elongated forms of nanoparticles are observed.

As can be seen in the Raman spectra presented in Figure 4, when the zinc acetate is used as the precursor, the as-synthesized ZnO NPs show the peaks corresponding to the hexagonal wurtzite structure of ZnO. These peaks are located at around 330 cm^−1^, 385 cm^−1^, 439 cm^−1^ and 585 cm^−1^ and attributed to, as follows: multiphonon modes, A_1_(TO) mode, E_2_^high^ mode and E_1_(LO) mode, respectively [26]. In the case of 4 LiOH/zinc acetate ZnO nanoparticles, the polar A_1_(TO) mode intensity is bigger, indicating the increase in the crystalline disorder. In contrast to these results, as it is shown in Figure 5, the Raman signal intensity of the 4 LiOH/zinc sulfate sample is low with no sharp peaks, which indicates that the crystalline quality of the nanoparticles synthesized with 4 LiOH/zinc sulfate is worse than that of the NPs synthesized with 4 KOH/zinc sulfate.

### 3.3. TEM Analysis

From the TEM images in Figure 3, it can be observed that when 4 KOH is used as the alkaline hydroxide, the resulting ZnO nanoparticles are in the form of needles, while when 4 LiOH is used, nanoplatelets are formed. Nevertheless, the mean crystalline domain size shrinks from 20 nm (for KOH) to 13 nm (for LiOH) when acetate is used, whereas it increases from 20 nm (for KOH) to 27 nm (for LiOH) in the sulfate case. As can be seen from the HRTEM images (Figure 6), the diameter of the needles corresponds to the size of crystalline domain deduced from the analysis of XRD results. It shows that the ions of Li^+^ are very important factors in the control of the ZnO NPs’ morphology, whereas the size of the crystalline domain is determined by the nature of the precursor. This is perfectly consistent with the XRD results in Figure 2. However, neither the needles in Figure 3a,c nor the nanoplatelets in Figure 3b,d are monodomain crystals (Figure 6). Usually, the growth rate is the highest in the Zn-terminated (0001) plane, resulting in the nanoneedle shape [27]. As Li^+^ is added, the adsorption of Li^+^ slows down the growth rate along the [001] axis and accelerates that along the [00-1] and [1010] axes through changing the surface charge and the attachment of zinc-containing species [28], resulting in the ZnO NPs in the form of nanoplatelets. Thus, the size variation is determined by the nature of the precursor and of the hydroxide. The size reduction in the presence of LiOH compared to KOH and NaOH with Zn acetate has been reported previously [19] and is explained by the different kinetic growth between alkaline ions (K^+^, Na^+^ and Li^+^). The K^+^ and Na^+^ ions interact weakly with the surface of a nanoparticle. They can easily be replaced with OH^−^ groups, leading to the attraction of Zn^2+^ ions and the nanoparticle growth progresses gradually, almost independently of the hydroxide type (KOH or NaOH). On the contrary, the Li^+^ ion with a smaller ionic radius binds stronger to the NP surface, which results in the repulsion of Zn^2+^ ions and attraction of the precursor anions (acetate or sulfate ions). The acetate ion is known to have a strong stabilizing influence on the nanoparticles [29]. The ZnO NPs grown in this case remain small and stabilized in their final form. On the contrary, sulfate ions have a lower stabilizing power. The growth of the NPs can proceed for a longer time, resulting in larger particles.

Two equivalents of KOH reacting with zinc acetate gave mainly oligomeric Zn(OH)_x_(OAc)_y_ species, while two equivalents of LiOH resulted in ZnO precipitate. The use of sulfate reacting with two equivalents of LiOH produced a mix of wurtzite ZnO and wulfingite Zn(OH)_2_ with a respective ratio of 95:5. The rough estimation of the ZnO crystalline domain sizes using Scherrer’s formula (1) is also given in Table 1. FFT of the HRTEM image of KOH/Zn acetate exhibits a diffraction pattern (see Figure 6b) which looks like a powder one.

### 3.4. Optical Properties of ZnO Nanoparticles

The influence of studied ZnO nanoparticle powder composition on the photoluminescence properties and their efficiencies was investigated. Figure 7 and Figure 8 demonstrate the photoluminescence spectra of ZnO NPs which were synthesized using precursors of zinc acetate and zinc sulfate, respectively. Additionally, various equivalents of KOH and LiOH were used. One can notice that a broad and strong visible emission peak, occasionally complemented with a weak emission in the ultraviolet, dominated all the presented spectra. The intensive visible emission peak was analyzed with Gaussian fitting. The results are shown in Figure 7 and Figure 8. The UV and the visible (VIS) emissions were fitted with the minimum of the Gaussian peaks and within the ranges attributed to different defects, as presented in Table 3. The maxima and the integrated area under each peak were used to calculate the percentage of the presence of the given defect in the synthesized NPs. The results are plotted in the form of pie charts presented in Figure 9a–c and Figure 10a–c. There exists a continuum of defects within the bandgap of ZnO NPs which are possible, as well as the complexes between them; that is why the attribution of the emission peaks can be very challenging. For the peaks above 720 nm, we attribute them to a point defect such as O_i_ or other defect complexes present in the studied nanoparticles. The defects present in each sample along with their relative percentage are summarized in Table 4. We noticed that the visible emission band of both investigated precursors are characterized by the sub-bands, which can be decomposed in several emission lines, going from blue (or even UV in the case of 4 KOH/Zn acetate) via yellow to NIR. This wide emission is the reason for the application of ZnO NPs-based layers as down-converters in thin film solar cells [8]. The presence of the excitonic emission in the case of the 4 KOH/Zn acetate is the proof of its good crystalline quality, which is also confirmed by the XRD. The XRD peaks are of highest intensity for the 4 KOH/Zn acetate sample. Raman spectra also confirm the better crystalline quality of the samples synthesized with the KOH than with LiOH.

It can be observed that the quantum yield of the photoluminescence increases when the amount of LiOH is higher. An analogous trend is shown for both precursors with which ZnO NPs were synthesized (acetate and sulfate). Hence, the simple conclusion from this observation is that the presence of Li ions in the synthesis intensifies the concentration of the defects in the structure, which is at the origin of the emission of visible light. Furthermore, it can be noticed that this emission, caused by the defects, predominates over the excitonic one, and it also begins to compete positively with the recombination channels which are non-radiative.

Again, considering the ZnO nanoparticles synthesized with zinc acetate, it can be seen that with increased concentration of Li ions, the PL QY increases and the quantity of the defects responsible for the visible emission increases (see pie charts in Figure 9).

In the second investigated case, where ZnO NPs were produced using zinc sulfate, emission contributions are governed by the quantity of OH^−^ ions, in addition to the amount of ions of Li^+^. The oxygen vacancies (neutral or charged) are present in bigger quantities for the 4 KOH sample than for the 2 LiOH and 4 LiOH. When either KOH, or a low concentration of LiOH, was used for the NPs’ synthesis, lower values of the PL QY with zinc acetate than that with zinc sulfate are obtained. Nonetheless, at a high quantity of Li^+^, the PL QY escalates and is equal for both of the investigated precursors. As this high PL QY is observed for both samples, where Zn(OH)_2_ is present (see XRD spectra for 4 LiOH/Zn acetate and 4 LiOH/Zn sulfate), we can suppose that there is a connection with the species of stable OH surface states which can be supported on the surface of ZnO in the form of Zn(OH)_2_. In this case, a radiative decay path would involve a transition from OH states to deep-level states [30,31], especially if the preparation method includes a hydrated form of zinc acetate.

Li^+^ ions are recognized to behave as hole scavengers when they are placed in the interstitial sites [32] and are at the origin of the green emission near 570 nm. Conversely, they become deep acceptors when they are in substitutional position and replace Zn atoms [16]. In some cases (2 LiOH/Zn sulfate and 4 LiOH/Zn acetate), we can suppose the presence of the zinc interstitials, as suggested by the Gaussian fits (see pie charts in Figure 9a,c and Figure 10b). As already mentioned, it was not possible to detect any Li by the NRA, indicating that its atomic concentration in the ZnO nanoparticles is smaller than the 0.5% NRA detection limit. In addition, for both precursors, the content of the defects such as V_Zn_ and V_O_ (neutral or charged) does not follow the variation in the Li^+^ concentration, as shown in Table 4. The syntheses have been performed at ambient conditions in water, i.e., corresponding to the O-rich milieu, in which V_Zn_ and O_i_ have the lowest formation energy and thus are the most probable defects [33]. Since the sub-bandgap absorption under 532 nm excitation and the ZnO red emission were attributed to a single defect—V_Zn_ [34]—the PL measurement using a 532 nm laser on co-precipitation 4 LiOH/Zn acetate synthesized ZnO NPs was performed. These NPs yield the optimum PL QY (see Table 4). The PL spectrum is shown in Figure 11. It can be noticed that the ZnO NPs absorb the photons below their bandgap energy and re-emit at about 630 nm. Hence, the presence of V_Zn_ in the samples of ZnO NPs is confirmed. However, V_Zn_ and its related defect complexes are more stable on the non-polar (100) planes [35,36]. For both precursors, the Li^+^ ions inhibit the growth along the [001] direction, as mentioned previously, while they increase that along the [100] axis. Consequently, with the use of Li^+^ ions, the fraction of the non-polar (100) plane should decrease as well as the V_Zn_ concentration, leading to a decrease in the PL QY. Since this is not what we observe, it can be inferred that the V_Zn_ is not the only defect at the origin of the visible emission. The origin of the visible emission in ZnO nanoparticles is still a subject of controversy [37,38], though there a consensus emerged over the fact that the green emission (at around 2.27 eV) results from the presence of the singly ionized oxygen vacancies or their complexes [1]. The question is whether these transitions are the ones between electrons in the conduction band and holes on deep defects or a hole in the valence band and the electron trapped on the oxygen vacancy. Whatever the exact phenomenon, the presence of the oxygen vacancies is intimately related to the green emission in our ZnO NPs.

Li et al. [37] have attributed both yellow and red emission to O_i_, located in the bulk and on the surface, respectively. Thus, the defect responsible for the visible emission could be the V_Zn_-O_i_ complex mediated by Li ions. When acetate is used as precursor, the I_1_/I_2_ ratio does not vary with the nature of the hydroxide, and their respective equivalent quantities change, indicating that the ration between O_i_ at the surface and in the bulk remains rather constant. This can be related to the strong stabilizing effect of the acetate. In the meantime, the increase in the Li^+^ concentration increases the PL QY, probably by creating more of the aforementioned defects.

In the case of the zinc sulfate precursor, the situation is different. Raman spectroscopy shows that the crystal network is strongly influenced by LiOH. Moreover, XRD patterns show an increase in the mean size of the crystalline domain of the zinc sulfate-LiOH synthesized ZnO NPs. In spite of the relatively large size, the radiative defects are more often present. Henceforward, the PL QY increases as the Li^+^ content grows. The introduction of LiOH in the synthesis milieu of ZnO nanoparticles has already been reported in the literature [7], and the control of the pH of the solution by the presence of Li^+^ was put forward. Keeping the basic pH of the reaction milieu results in the oxidizing character of the solution. Since the sulfate has a weaker stabilizing power than the acetate, the ZnO NPs synthesized with sulfate are much more sensitive to the OH^−^ concentration, which, in turn, influences the O_i_ defect creation and the diffusion of these defects [38], resulting in a higher content of these defects for ZnO NPs synthesized with sulfate, if we compare high PL QY samples (content of O_i_ of 13.4% for 4 LiOH/Zn acetate vs. 33.7% for 4 LiOH/Zn sulfate). It can be concluded that the nanoparticle size has an influence (the smaller the better), but also that the fraction of the not-completely crystallized ZnO NPs facets have a substantial impact on the intensity of their yellow-red luminescence.

The results of the photoluminescence excitation study of the co-precipitation synthesized (4 LiOH/Zn acetate) ZnO NPs are presented in Figure 12. The PLE spectrum has a high intensity in the spectral range from 350 to 370 nm. It demonstrates that the visible luminescence of the studied ZnO NPs originates from the photons absorbed in the range between 350 and 370 nm. It also confirms that the nanoparticles absorb the UV light and re-emit photons at longer wavelengths via the downshifting process.

## 4. Conclusions

With the help of the co-precipitation technique, ZnO NPs were successfully synthesized. The synthesis was realized with two different zinc precursors: Zn acetate and Zn sulfate with two various alkaline hydroxides (KOH and LiOH), diluted in water or in ethanol. By these experiments, it was confirmed that the size of the ZnO NPs synthesized in ethanol is usually smaller than the size of NPs synthesized in water using KOH, even though the PL QY is the same and very low, indicating that the optical behavior is ruled by non-radiative defects. Due to the high abundance and non-toxicity of water, and as the solvent did not influence the PL QY, water was applied as solvent for further syntheses. Consequently, the replacement of KOH with LiOH in the synthesis process, with an LiOH concentration increase, results in the NPs’ size decrease with Zn acetate, while it increases when ZnO sulfate is used. This phenomenon may be explained by the stronger ability to protect the primary nanoparticle embryos from aggregating in the case of acetate rather than sulfate. In both types of ZnO NPs (synthesized with acetate or sulfate), the nanoparticle morphology shifts from nanoneedles to nanoplatelets while KOH is replaced with LiOH within the synthesis process. It turns out that in the case of the potassium hydroxide, the nanoparticles grow preferentially in the (002) direction, which can also be noticed in TEM images (Figure 3b,d). At the same time, the intensity of the visible light emission is magnified with the increase in the LiOH concentration, with the PL QY reaching 13%, and it is clear that the presence of Zn(OH)_2_ is favorable for this high PL QY. We observe that Zn(OH)_2_ is beneficial for high PL QY regardless of the salt used. To the best of our knowledge, studies performed on the Li or K-doped ZnO NPs synthesized by the co-precipitation method rarely concerned the PL QY [17,39,40,41,42,43]. Considering very low concentration (below 0.5%) of the doping atoms, the value of 13% of the PL QY can be considered as comparable (or reasonably, even higher) to the value to the ZnO NPs doped with Li, which exhibited around 15% of the external quantum efficiency of the photoluminescence [44], especially given that in our case we measured the internal PL quantum efficiency. This visible emission should originate from the presence of intrinsic defects rather than exclusively from the Li dopant. This enhancement is directly related to the increase in the concentration of the intrinsic defects. For the two alkaline hydroxide precursors used, the mechanisms of the amplification of the defect-related emission are different. In the case of acetate, it is exclusively influenced by Li^+^ ion concentration, whereas for sulfate, it is also influenced by the OH^−^ concentration. It clearly confirms that acetate presents stronger surface stabilization potential for the ZnO NPs’ growth than sulfate.

## Figures and Tables

**Figure 1 materials-16-05400-f001:**
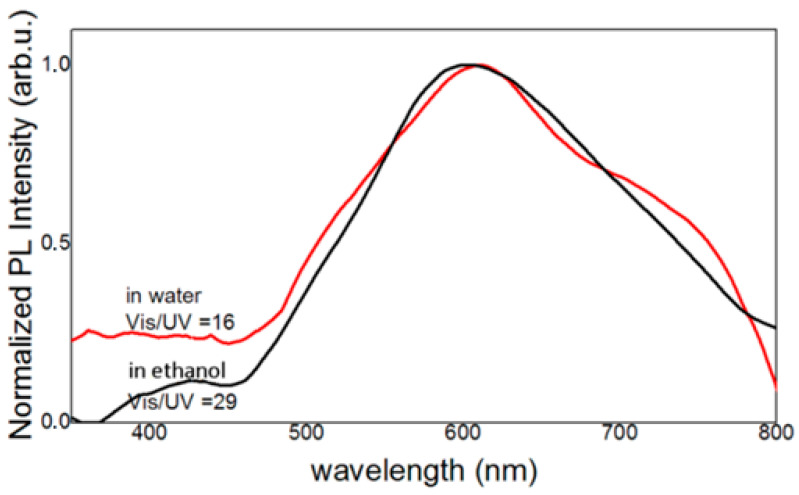
Photoluminescent emission spectra of ZnO nanoparticles in powders synthesized by the co-precipitation of the zinc acetate with KOH (4 KOH/Zn acetate) in ethanol (black curve) and in water (red curve).

**Figure 2 materials-16-05400-f002:**
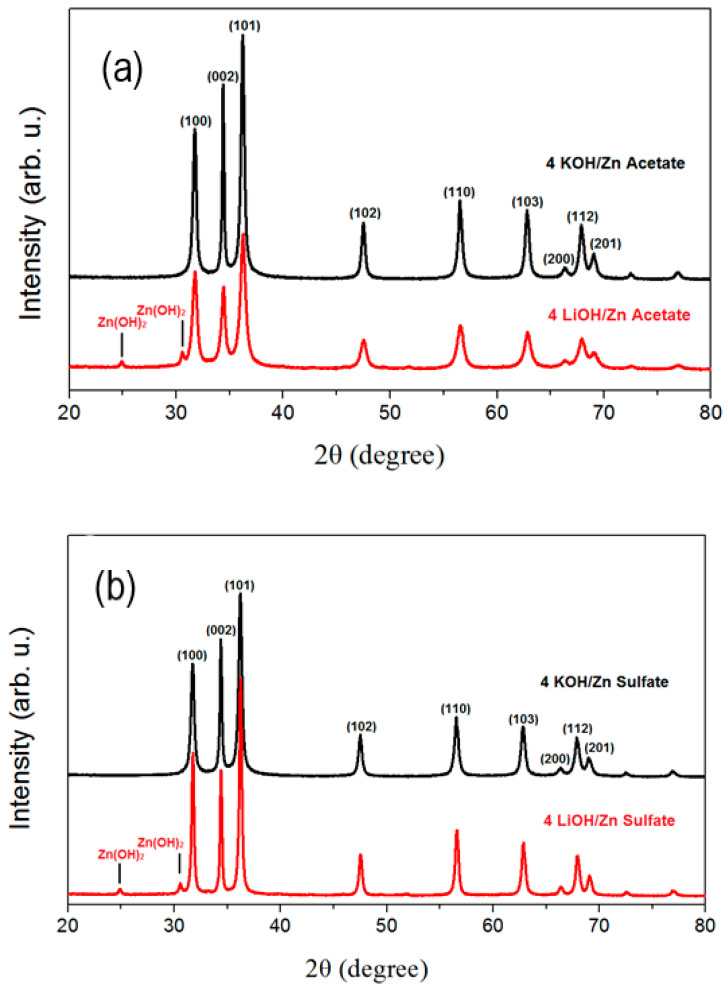
XRD patterns of co-precipitation synthesized ZnO powders: (**a**) 4 KOH/Zn acetate (top curve) and 4 LiOH/Zn acetate (bottom curve); (**b**) 4 KOH/zinc sulfate (top curve) and 4 LiOH and zinc sulfate (bottom curve). In both graphs, the curves were moved along the *y*-axis for clarity.

**Figure 3 materials-16-05400-f003:**
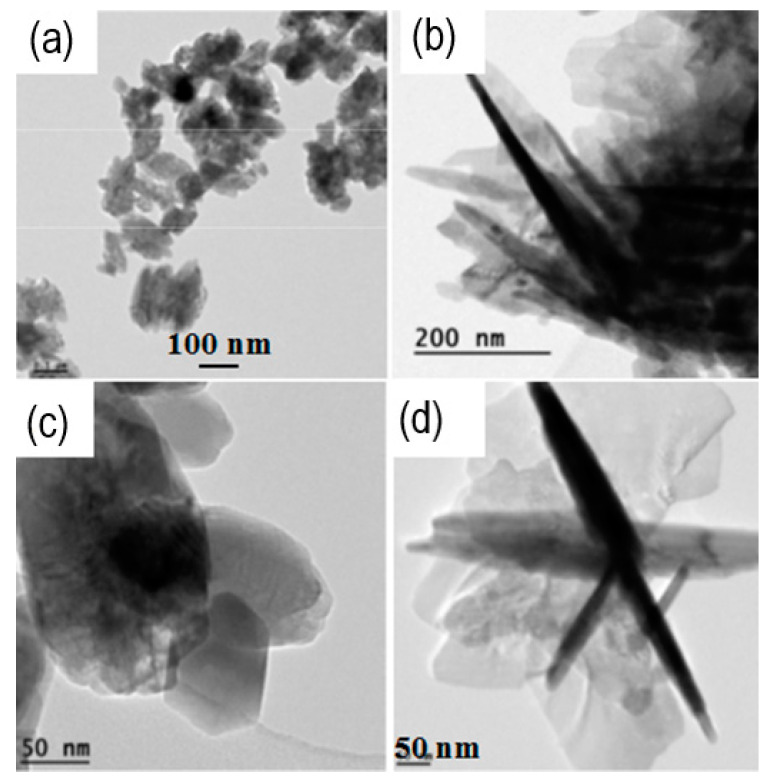
TEM images of the ZnO nanoparticle powders synthesized by the co-precipitation method of 4 LiOH/Zn acetate (**a**), 4 KOH/Zn acetate (**b**), 4 LiOH/Zn sulfate (**c**) and 4 KOH/Zn sulfate (**d**).

**Figure 4 materials-16-05400-f004:**
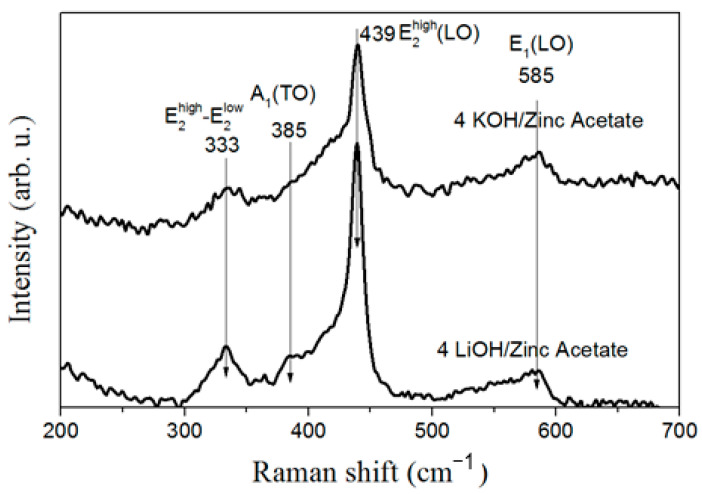
Raman spectra of ZnO nanoparticle powders synthesized by the co-precipitation of zinc acetate with KOH (4 KOH/Zn acetate) (top curve) and LiOH (4 LiOH/Zn acetate) (bottom curve). The curves were translated along the *y*-axis for clarity.

**Figure 5 materials-16-05400-f005:**
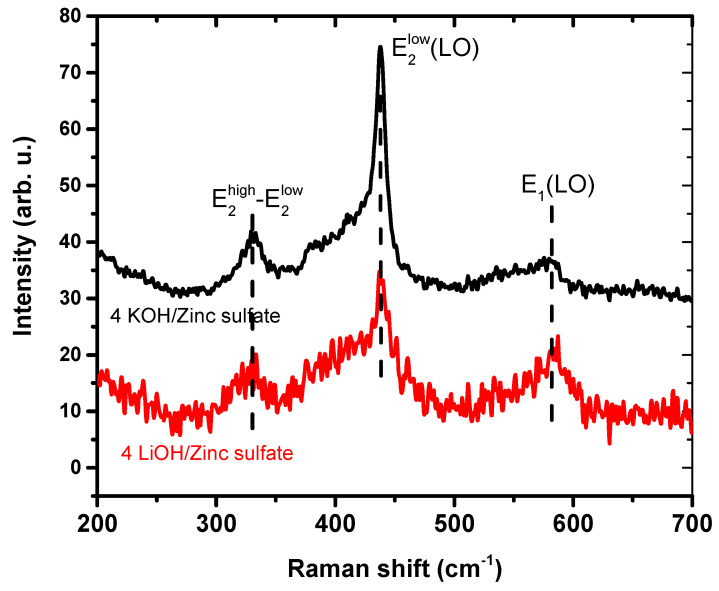
Raman spectra of zinc oxide (ZnO) nanoparticle powders synthesized using the method of the co-precipitation of zinc sulfate and KOH (4 KOH/Zn sulfate) (top curve) and the co-precipitation of zinc sulfate with LiOH (4 LiOH/Zn sulfate) (bottom curve). Both curves were moved alongside the *y*-axis in order to make the graph clearer. Peaks observed are $E_{2}^{\mathrm{high}}-E_{2}^{\mathrm{low}}$ at 332 cm^−1^, $E_{2}^{\mathrm{low}}\left(\mathrm{LO}\right)$ at 438 cm^−1^ and $E_{1}\left(\mathrm{LO}\right)$ at 584 cm^−1^.

**Figure 6 materials-16-05400-f006:**
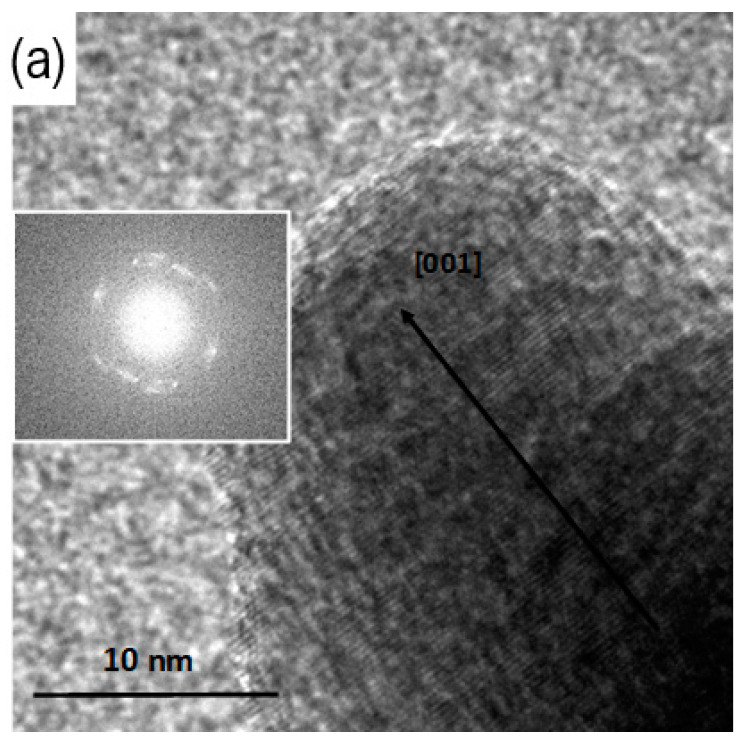
HRTEM images of the ZnO nanostructures synthesized by the co-precipitation of the zinc acetate with: 4 KOH (**a**), 4 LiOH (**b**) in water. A staking default is observable at the bottom of the image (see arrow). Insets are the corresponding FFT images.

**Figure 7 materials-16-05400-f007:**
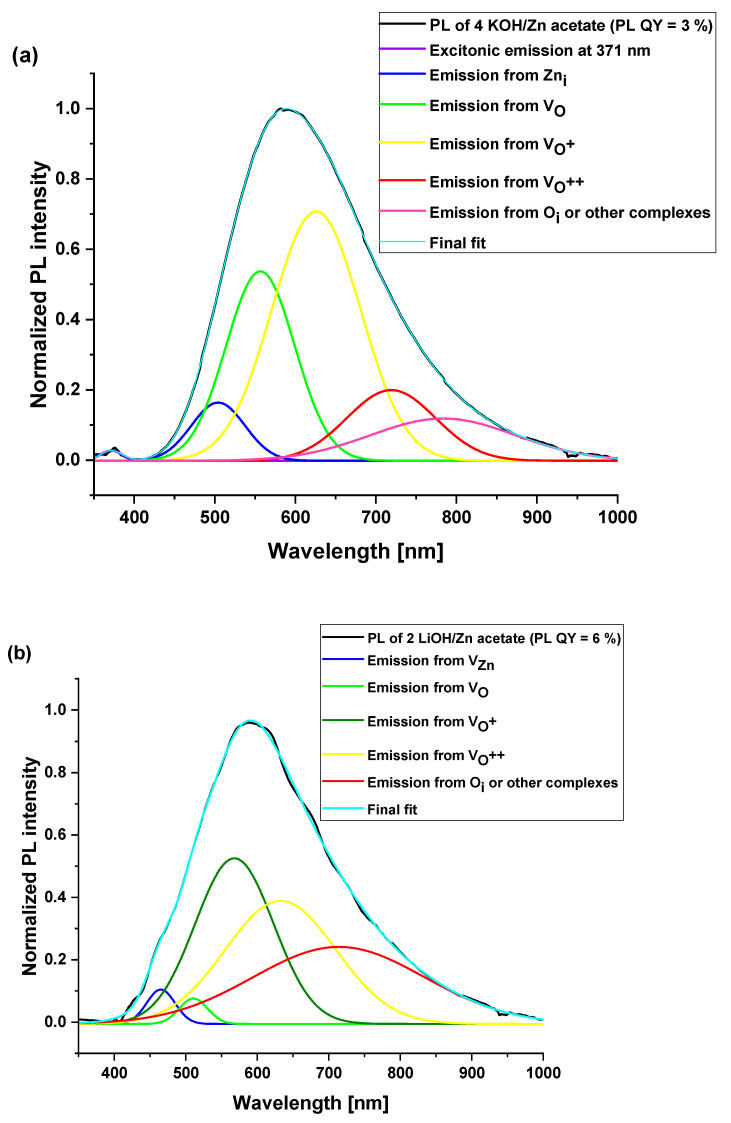
Normalized PL spectra of ZnO nanoparticle powders synthesized by the co-precipitation method of (**a**) zinc acetate with KOH (4 KOH/Zn acetate), (**b**) LiOH (2 LiOH/Zn acetate), and (**c**) 4 LiOH/Zn acetate, bottom curve. For each PL spectrum, the experimental curve is shown in black line and the color curves are the Gaussian fits of the experimental emission peaks.

**Figure 8 materials-16-05400-f008:**
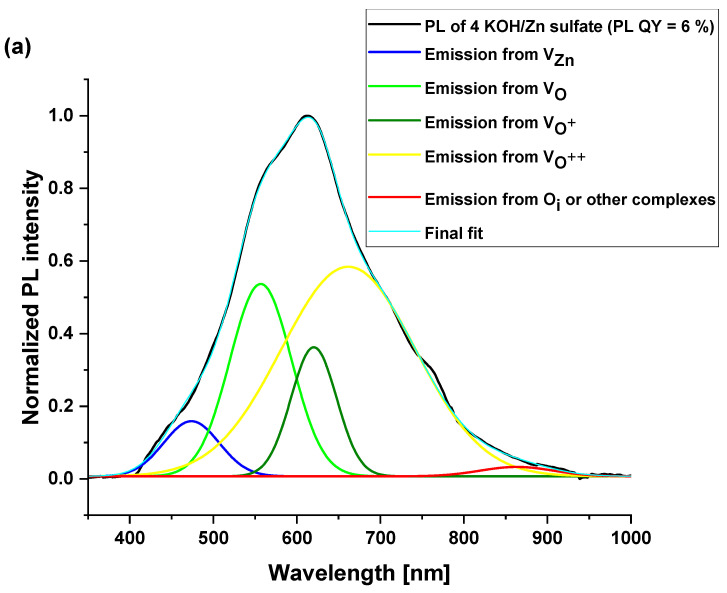
Normalized PL spectra of co-precipitation ZnO nanoparticle powders synthesized using (**a**) the zinc sulfate with KOH (4 KOH/Zn sulfate), (**b**) LiOH (2 LiOH/Zn sulfate), and (**c**) 4 LiOH/Zn sulfate. For each PL spectrum, the experimental curve is shown in black line and the color curves are the Gaussian fits of the experimental emission peaks.

**Figure 9 materials-16-05400-f009:**
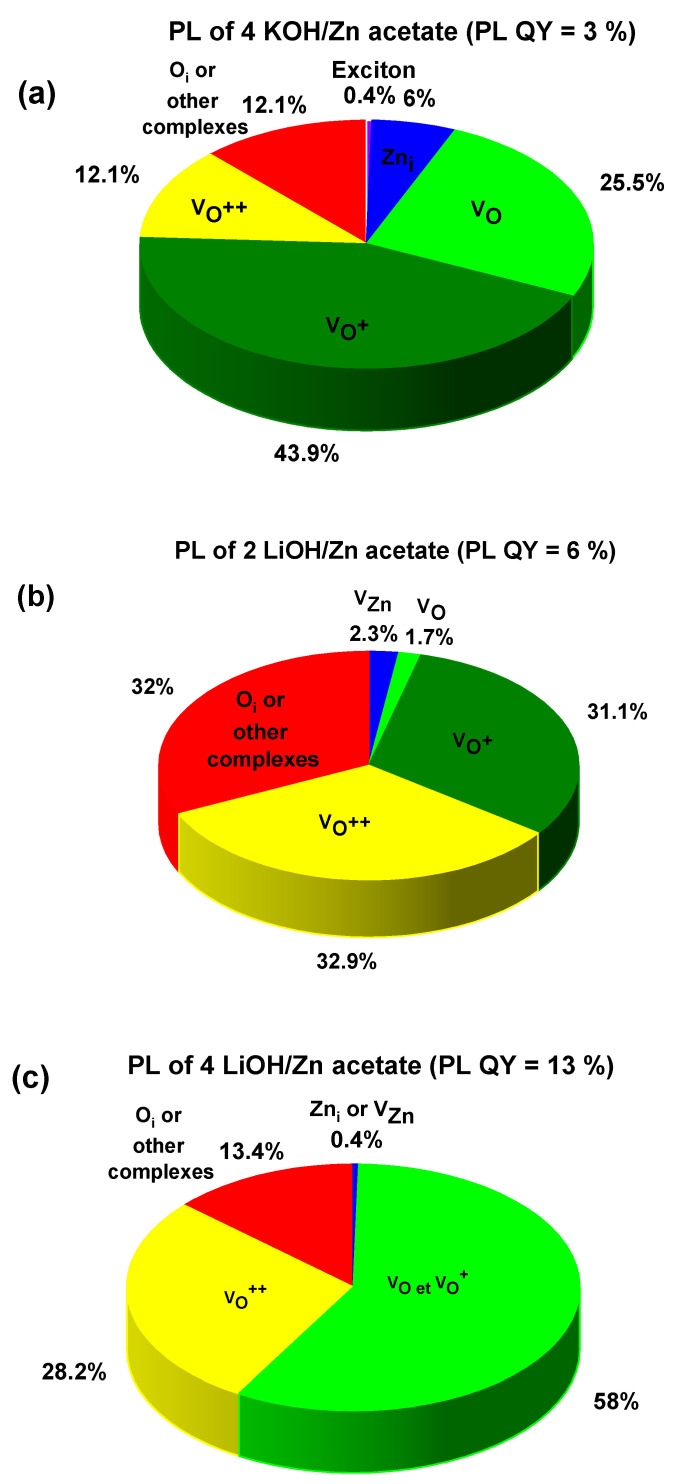
Pie charts of the relative percentage of defects for the PL spectra presented in the Figure 7.

**Figure 10 materials-16-05400-f010:**
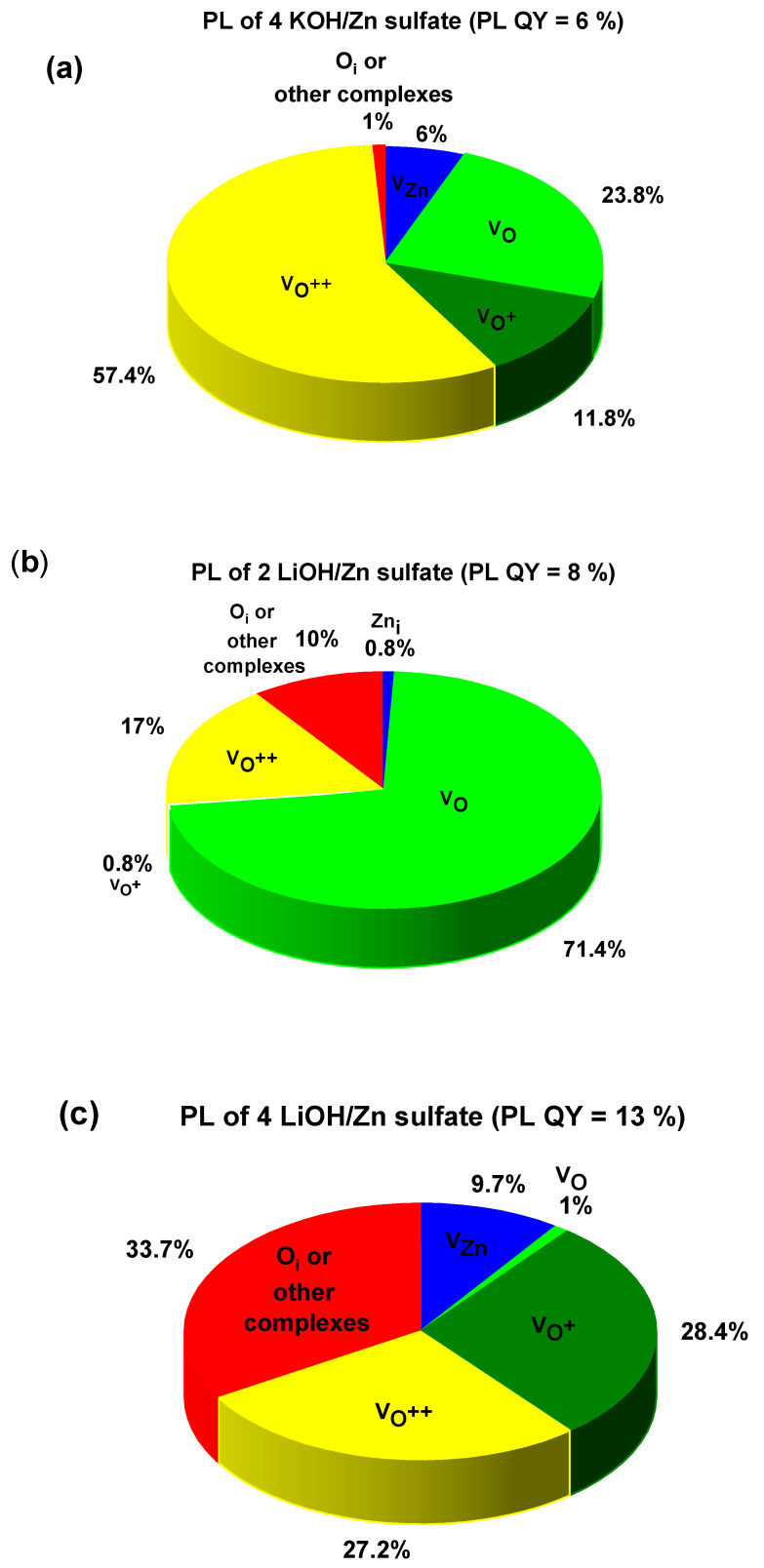
Pie charts of the relative percentage of defects for the PL spectra presented in the Figure 8.

**Figure 11 materials-16-05400-f011:**
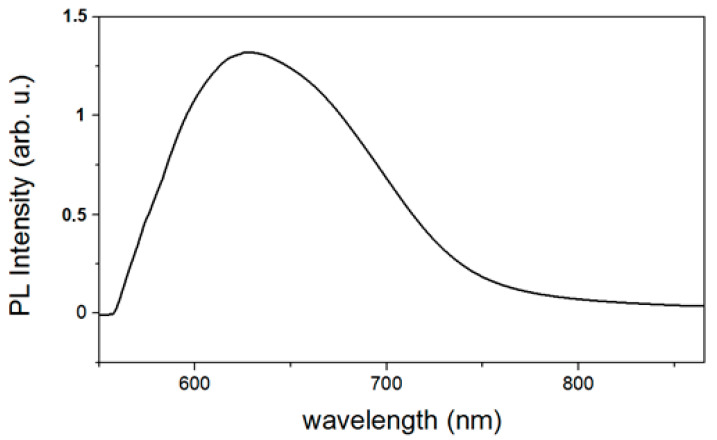
PL emission spectrum of co-precipitation synthesized (4 LiOH/Zn acetate) ZnO NPs powder under the excitation of 532 nm and which has the optimum PL QY.

**Figure 12 materials-16-05400-f012:**
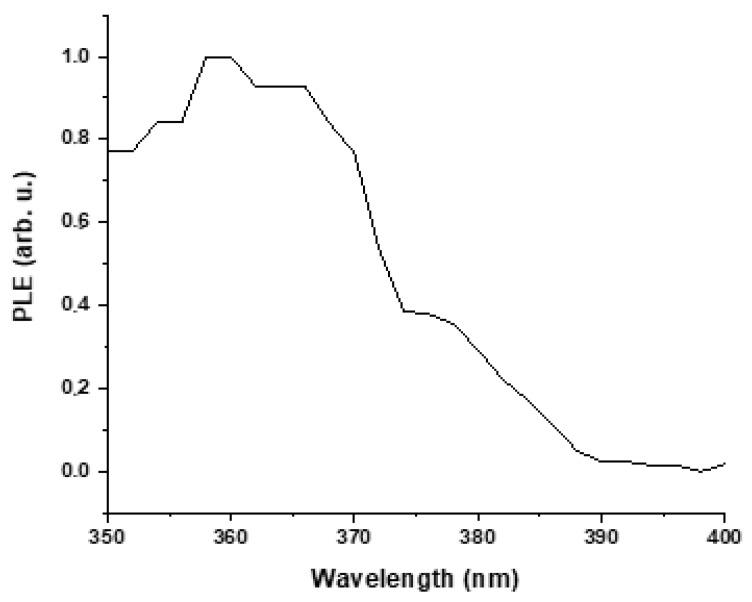
PLE emission spectrum of co-precipitation synthesized (4 LiOH/Zn acetate) ZnO NPs powder which has the optimum PL QY.

**Table 1 materials-16-05400-t001:** Estimated ZnO NPs’ crystalline domain sizes deduced from the XRD diffractograms presented in Figure 2.

Precursor	Hydroxide		
	KOH	LiOH	
	4 eq.	2 eq.	4 eq.
Zinc acetate	20 nm	21 nm	13 nm
Zinc sulfate	20 nm	23 nm	27 nm

**Table 2 materials-16-05400-t002:** FWHM of (101) and (002) peaks from the XRD diffractograms presented in Figure 1 and crystalline domain sizes deduced from the Scherrer formula.

Precursors	FWHM of (101) Peak and Resulting Crystalline Domain Size	FWHM of (002) Peak and Resulting Crystalline Domain Size
4 LiOH/Zn acetate	0.63° (13 nm)	0.57° (15 nm)
4 KOH/Zn acetate	0.44° (20 nm) *	0.36° (24 nm)
4 LiOH/Zn sulfate	0.32° (27 nm)	0.31° (28 nm)
4 KOH/Zn sulfate	0.44° (20 nm) *	0.33° (26 nm)

* the value of the crystal domain size obtained from the calculation for the (101) peak was taken as the crystalline domain size in the text.

**Table 3 materials-16-05400-t003:** Various defects present in the studied samples [21]. The peak ranges are used in the Gaussian fitting of the PL spectra of Figure 7 and Figure 8.

Surface Defects	Peak Range (nm)
V_Zn_	470–520
V_O_	520–570
V_O_^+^	570–620
V_O_^++^	620–670
O_i_	670–720

**Table 4 materials-16-05400-t004:** Summary of peak positions I_1_, I_2_, I_3_, I_4_ and I_5_ and (in brackets), the area under Gaussian peaks (in %) obtained from the fits of the PL spectra of the synthesized ZnO NPs with Zn acetate and Zn sulfate.

Sample Synthesis Condition	I_1_ (nm)	I_2_ (nm)	I_3_ (nm)	I_4_ (nm)	I_5_ (nm)	PL QY
4 KOH/ Zn acetate	371 (0.4) (exciton) and 504 (6)	556 (25.5)	625 (43.9)	719 (12.1)	783 (12.1)	3 ± 1%
2 LiOH/ Zn acetate	465 (2.3)	510 (1.7)	567 (31.1)	632 (32.9)	714 (32)	6 ± 1%
4 LiOH/ Zn acetate	443 (0.4)	566 (58)	566 *	651 (28.2)	743 (13.4)	13 ± 1%
4 KOH/ Zn sulfate	473 (6)	556 (23.8)	620 (11.8)	661 (57.4)	863 (1)	6 ± 1%
2 LiOH/ Zn sulfate	401 (0.8)	584 (71.4)	611 (0.8)	709 (17)	811 (10)	8 ± 1%
4 LiOH/Zn sulfate	502 (9.7)	530 (1)	578 (28.4)	649 (27.2)	693 (33.7)	13 ± 1%

* No area is indicated as the peaks of V_O_ and V_O_^++^ merged in this case.

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
