# Peer review of "Improved Visible Emission from ZnO Nanoparticles Synthesized via the Co-Precipitation Method"

_materials, 2023, doi:10.3390/ma16155400_

Round 1

Reviewer 1 Report

In this work, the author propose to synthesize ZnO nanoparticles via the co-precipitation method. Furthermore, the influence of the synthesis conditions on the nature of the precursor, the type and amount of alkaline hydroxide as well as the effect of the reaction solvent on the crystalline quality of the nanoparticle, size, morphology and optical properties were studied.

However, there are still some points that the author must resolve before acceptance. The detailed list of comments is presented below:

1. The introduction is well organized, but also taking into account that ZnO is a well-studied material, I recommend the authors to present some results from the literature that study the effects of the synthesis method on optical properties, particularly the emission centers (defects) , since the authors state that this is rarely studied. For example, there are some recent works that correlate the effect of pH on structural, magnetic and optical properties. https://doi.org/10.1016/j.ceramint.2022.09.196

2. Materials and Methods.

Preparation of ZnO nanoparticles.

I recommend that authors include a description of the synthesis of nanoparticles. Since it is an important part of the research.

3. Structural studies. I recommend estimating structural parameters derived from the Williamson-Hall analysis, the structural parameters, lattice constant, lattice stress and dislocation density. It can improve the discussion of the work and correlate with the optical properties and the effect of the solvent for example.

4. Optical properties. The photoluminescence (PL) of ZnO nanoparticles exhibits near-band edge (NBE) and visible luminescence associated with exciton transitions and defect emissions, respectively. The emission centers in the visible region are dependent on the synthesis technique and, therefore, on vacancies, surface defects and morphology. The understanding of the visible PL mechanism of ZnO is still an open question. Deep Level Emissions (DLE) denote the levels allowed within the bandgap of the semiconductor that produces transitions with energy in the visible range of the spectrum. Different types of defects can exist at the same time, which results in a broad-band emission. There are many works in the literature reporting the observation of different emissions.

The deconvolution of the photoluminescence spectrum fitting the Gausian distribution function must be revised and corrected. The deconvolution of the spectrum must follow the Rayleigh criterion. I recommend authors review the articles by K. Punia et. al. how to perform deconvolution. These results are very important in this research.

https://doi.org/10.1016/j.jallcom.2021.159142

Author Response

We thank the Rewiever for the constructive comments.

  1. We added the references to the introduction, as suggested by the Rewiever. These are the references number 21 and 22.
  2. The description of the sythesis was added to the article. It was an omission during the submission process, as this part of the article was supposed to be there from the very beginning. We are sorry about it.
  3. We agree that the more precise estimation according to the Williamson-Hall analysis of the nanoparticule structure, including the presence of the dislocations or even the local strain could improve the discussion, but our main message concerns more the synthesis parameters necessary for obtaining the highest PL QY.

    Dislocations might have impact on the quantum yield, leading to PL quenching. On the other hand, strain may shift the luminescence features. Therefore, such an analysis is very relevant and we will be in the scope of the forthcoming articles since it deserves specific attention.

    4. We retreated the PL spectra, with more careful attention to the noise level and we performed the deconvolution with different Gaussian peaks, as suggested. We also presented the results in form of pie charts.

Reviewer 2 Report

Regarding manuscript materials-2497685, entitled „Improved visible emission from ZnO nanoparticles synthesized via the co-precipitation method“ by authors A. Apostoluk, Y. Zhu, P. Gautier, A. Valette, J. M. Bluet, T. Cornier, B. Masenelli and S. Daniele.

The presented manuscript is focused on the research of the visible emission fron ZnO NPs, as a function of two zinc precursors, two solvents and the addition of LiOH or KOH. The prepared samples are characterized by XRD, Raman spectroscopy, TEM, nuclear reaction analysis and photoluminescence. The presented study and the obtained data is well presented.

The paper can be accepted after minor revision:

1. The preparation of ZnO NPs is not given in the Materials and Methods. Actually, there is subtitle „Preparation of ZnO nanoparticles“, but the section presents only characterization methods.

2. XRD study: FWHM of (002) lines can be given also in Table . Which XRD lines are used for estimation of the crystallite sizes given in Table 1?

3. Conclusions can be written in a clearer style. It can give the effect of KOH and LIOH when Zn acetate is used and then for the other type of precursor. Depending on the whole performed research, including crystallization behavior, morphological features and luminescence quantum efficiency, it will be good to be clearly stated the optimal technological parameters to obtain a high PL QY.

Author Response

We thank the Rewiever for the constructive comments.

  1. The description of the synthesis was added to the article. It was an omission during the submission process, as this part of the article was supposed to be there from the very beginning. We are sorry about it.
  2. We added a Table 2 in which we put the values of FWHM of (101) and (002) peaks from the XRD diffractograms presented in Figure 1 and crystalline domain sizes deduced from the Scherrer’s formula for both peaks. It turns out that in the case of the potassium hydroxide, the nanoparticles grow preferentially in the (002) direction , which can also be noticed in TEM images (Figure 5 b) and d)), independently on the salt used.
  3. We have reformulated the conclusions, hoping they are clearer.

Round 2

Reviewer 1 Report

The manuscript can be considered for publication.